# A benchmark of categorical encoders
# for binary classification

Federico Matteucci[1], Vadim Arzamasov[1], and Klemens Böhm[1]

[1]Karlsruhe Institute of Technology
{federico.matteucci, vadim.arzamasov, klemens.boehm}@kit.edu

## Abstract

Categorical encoders transform categorical features into numerical representations that are indispensable for a wide range of machine learning models. Existing encoder benchmark studies lack generalizability because of their limited choice of **1.** encoders, **2.** experimental factors, and **3.** datasets. Additionally, inconsistencies arise from the adoption of varying aggregation strategies. This paper is the most comprehensive benchmark of categorical encoders to date, including an extensive evaluation of 32 configurations of encoders from diverse families, with 48 combinations of experimental factors, and on 50 datasets. The study shows the profound influence of dataset selection, experimental factors, and aggregation strategies on the benchmark's conclusions — aspects disregarded in previous encoder benchmarks. Our code is available at https://github.com/DrCohomology/EncoderBenchmarking.

## 1 Introduction

Learning from categorical data poses additional challenges compared to numerical data, due to a lack of inherent structure such as order, distance, or kernel. The conventional solution is to transform categorical attributes into a numerical form, i.e., *encode* them, before feeding them to a downstream Machine Learning (ML) model. Various encoders have been proposed, followed by several benchmark studies. However, their combined results remain inconclusive, as we now describe.

Many factors impact the generalizability [26] of a benchmark of encoders, including: **1.** the compared encoders, **2.** the number of datasets, **3.** the quality metrics, **4.** the ML models used, and **5.** the tuning strategy. We also hypothesize that **6.** the *aggregation strategy* used to summarize the results of multiple experiments may affect the conclusions of a study. Existing encoder benchmarks, reviewed in Section 2, only partially control for these factors. First, none of these studies uses more than 15 datasets of a given type (regression or classification). Second, despite these studies collectively covering a substantial number of encoders, they often focus on specific encoder families, resulting in comparison gaps between the best encoders. For instance, the best-performing encoders from [28] (Cross-Validated GLMM) and [44] (Mean-Target) have not been studied together yet. Third, the results of existing studies are often not comparable due to variations in the selected quality metrics. For instance, [28] measures quality with ROC AUC, [4] with average precision, and [41] with accuracy. Fourth, existing studies tune ML models in different ways, yielding incompatible evaluations. For instance, [28, 41] do not tune, while [4, 5, 8, 44] tune but do not specify if they tune the ML model on encoded data or if they tune the entire ML pipeline. Last, no benchmark study of categorical encoders explores the impact of aggregation strategies, which is substantial according to our experiments. For instance, [5] ranks the encoders by average ranking across all datasets, while [28] computes the median ranking with Kemeny-Young aggregation [46].

This study offers a taxonomy and a comprehensive experimental comparison of encoders for binary classification, taking into account the factors just mentioned. In particular, we consider: **1.** 32 encoder configurations, including all of the best-performing ones from the literature and three novel encoders; **2.** 50 datasets for binary classification; **3.** four quality metrics; **4.** five widely used ML models; **5.** three tuning strategies; **6.** 10 aggregation strategies gathered from existing categorical encoder benchmarks and from benchmarking methodology studies [27, 9]. This allows us to provide novel insights into the sensitivity of experimental results to experimental factors. In particular, we demonstrate how replicability [26] may not be ensured even for studies conducted on up to 25 datasets. For those combinations of experimental factors that show reproducible results, we isolate and recommended the best encoders.

Paper outline: Section 2 reviews existing works, Section 3 presents a taxonomy of encoder families, Section 4 describes the experimental setup, and Section 5 features the results.

Table 1: Related work on categorical encoders for binary classification.

| | | Ours | [28] | [4] | [8] | [5] | [44] | [41] |
|---|---|---|---|---|---|---|---|---|
| # Binary classification datasets | | 50 | 10 | 5 | 3 | 2 | 2 | 6 |
| # ML models | | 5 | 5 | 1 | 4 | 2 | 1 | 5 |
| Encoder family | Identifier | ✓ | ✓ | ✓ | ✓ | ✓ | ✓ | ✓ |
| | Frequency-based | ✓ | ✓ | | | | | |
| | Contrast | ✓ | | | | | | ✓ |
| | Similarity | ✓ | | ✓ | | ✓ | | |
| | Simple target | ✓ | ✓ | | ✓ | | | ✓ |
| | Binning | ✓ | ✓ | | | | | |
| | Smoothing | ✓ | ✓ | | | ✓ | | ✓ |
| | Data-constraining | ✓ | ✓ | | | | | ✓ |
| Quality metric | Precision-recall based | ✓ | | ✓ | ✓ | ✓ | | |
| | Balanced accuracy | ✓ | ✓ | | ✓ | | | |
| | Accuracy | ✓ | | | ✓ | | ✓ | ✓ |
| Tuning strategy | Full pipeline tuning | ✓ | | | | | ✓* | |
| | Model tuning | ✓ | | | ? | ✓ | | |
| | No tuning | ✓ | ✓ | | | | | ✓ |
| Aggregation strategy | Heuristic | ✓ | | | | ✓ | | |
| | Friedman-Nemenyi | ✓ | | | ✓ | | ✓ | |
| | Kemeny-Young | ✓ | ✓ | | | | | |

## 2  Related work

**Benchmarks of encoders.** We focus on binary classification tasks, as they offer a wider range of compatible encoders; indeed, we could conduct a deeper replicability analysis while maintaining the computation feasible. Table 1 summarizes the related work. The other benchmarks often consider few datasets and either do not tune the ML model or do not describe the tuning procedure. This limits their applicability and generalizability. Additionally, there are substantial differences in the experimental settings across articles, including the encoders considered, quality metrics employed, and aggregation strategies used to interpret results. Hence, the comparability of these findings is limited. For instance, [28] recommends a data-constraining encoder, [41] both data-constraining and contrast encoders, [5, 4] similarity encoders, [8] an identifier encoder, and [44] a simple target encoder. Other benchmarks of encoders are [36], which focuses on regression tasks and faces similar issues, and [30, 14, 20], that use only a single dataset.

**Analysis of benchmarks.** When designing our benchmark, we adhered to the best practices discussed in the literature on benchmark design and analysis. In particular, [27] studies how choices of experimental factors impact the experimental results and advocates for benchmarks that consider a large variety of factors. Similarly, [9] suggests guidelines to mitigate the inconsistencies in the choices of data and evaluation metric. Finally, [2] proposes a methodology to account for variance in the design choices (randomization of sources of variation) and post-processing of the experimental results (significant and meaningful improvements).

# 3 Taxonomy of encoders

This section presents the essential terminology and discusses the considered encoders and their corresponding families. Appendix 7.1 provides formal and detailed descriptions of the encoders.

## 3.1 Notation and terminology

Consider a tabular dataset with target $y$ taking values in $\{0, 1\}$, and let $\mathbf{A}$ be one of its attributes (columns). $\mathbf{A}$ is *categorical* if it represents qualitative properties and takes values in a finite domain $\Omega_A$. Each $\omega \in \Omega_A$ is a *level* of $\mathbf{A}$. Categorical attributes do not support arithmetic operations like addition or multiplication, and their comparison is not based on arithmetic relations. An *encoder* $E$ replaces a categorical attribute $\mathbf{A}$ with a set of numerical attributes, $E(\mathbf{A})$. We write $E(\Omega_A)$ to indicate the domain of $E(\mathbf{A})$. Encoders may encode different levels in $\mathbf{A}$ in the same way, or encode in different ways different occurrences in the dataset of the same level. Encoders are either *supervised* or *unsupervised*: Supervised encoders require a target column, while unsupervised encoders solely rely on $\mathbf{A}$. In what follows, $\mathbf{A}$ always denotes the categorical attribute to be encoded.

## 3.2 Unsupervised encoders

**Identifier encoders** assign a unique vector identifier to each level. The most recognized encoder is One-Hot (OH), the default encoder in most machine learning pipelines [11, 15]. One-Hot is both space-inefficient and ineffective [28, 4, 5]. Alternatives include Ordinal (Ord), which assigns a unique consecutive identifier to each level, and Binary (Bin), which splits the base-2 representation of Ord($\mathbf{A}$) into its digits.

**Frequency-based encoders** replace levels with some function of their frequency in the dataset. We use Count, which relies on absolute frequencies [28].

**Contrast encoders** encode levels into $(L-1)$-dimensional vectors so that the encodings of all levels sum up to $(0, \dots, 0)$ [41]. A constant intercept term, 1, is usually appended to the encoding of each level. Contrast encoders encode levels such that their coefficients represent the level's effect contrasted against a reference value. A common example is Sum, which contrasts against the target's average value.

**Similarity encoders** treat $\omega \in \Omega_A$ as strings and map them into a numeric space taking their similarity into account [5, 4]. These encoders are particularly useful for handling "dirty" categorical datasets that may contain typos and redundancies. One example is Min-Hash (MH), which decomposes each level into a set of $n$-grams, sequences of $n$ consecutive letters, and encodes to preserve the Jaccard similarity of the decompositions.

## 3.3 Supervised encoders

**Simple target encoders** encode levels with a function of the target. Prime examples are Mean-Target (MT) [7], which encodes with the conditional average $y$ given $\mathbf{A}$, and Weight of Evidence (WoE) [39], which encodes with the logit of MT($\mathbf{A}$). As Mean-Target can lead to severe overfitting [28, 31], it may benefit from regularization. The following families of encoders are regularization for Mean-Target.

We propose **Binning encoders**, that regularize MT by partitioning either $\Omega_A$ or MT($\Omega_A$) into bins. Pre-Binned MT (PBMT) partitions $\Omega_A$ to maximize the number of bins such that each bin's relative frequency exceeds a specified threshold, then encodes the binned attribute with MT. Discretized MT (DMT) partitions MT($\Omega_A$) into intervals of equal length, then encodes each level with the lower bound of the interval in which its MT encoding falls.

**Smoothing encoders** blend MT($\Omega_A$) with the overall average target. Notable examples are Mean-Estimate (ME) [22], which uses a weighted average of the two, and Generalized Linear Mixed Model encoder (GLMM) [28], which encodes with the coefficients of a generalized linear mixed model fitted on the data.

**Data-constraining encoders** regularize MT($\mathbf{A}$) by restricting the amount of data used to encode each occurrence of a level in the dataset. CatBoost (CB) [31] first randomly permutes the dataset's rows, then maps each occurrence of a level $\omega$ to the average target of its previous occurrences. Cross-Validated MT (CVMT) [28] splits the dataset into folds of equal size, then encodes each fold with an

MT trained on the other folds. We propose the BlowUp variant of CVMT, BUMT, which trains an MT on each fold and uses them to encode the whole dataset. Related variants are Cross-Validated GLMM (CVGLMM) [28] and its BlowUp version (BUGLMM).

## 4 Experimental design

As there is no intrinsic measure of an encoder's quality, we proxy the latter with the quality of an ML model trained on encoded data. This procedure is in line with the current literature on the topic, discussed in Section 2. Each experiment thus consists of the following steps. First, we fix a combination of factors: a dataset, an ML model, a quality metric, and a tuning strategy. Then, we partition the dataset using a 5-fold stratified cross-validation and pre-process the training folds by:

- imputing missing values with median for numerical and mode for categorical attributes;
- scaling the numerical attributes;
- encoding the categorical attributes.

If tuning is to be applied, we fine-tune the pipeline with nested cross-validation and output the average performance over the outer test folds. We used standard scikit-learn [29] procedures for scaling and missing values imputation.

We conducted experiments using Python 3.8 on an AMD EPYX 7551 machine with 32 cores and 128 GB RAM. We limit each evaluation to 100 minutes to handle the extensive workload. As described in Appendix 7.3.1, out of the 64000 cross-validated evaluations, 61812 finished on time without throwing errors. For the sensitivity, replicability, and encoder comparison analysis, we ignored the missing evaluations. We did so **1.** since there is no clearly superior imputation method, and **2.** to avoid introducing unnecessary variability in the analysis. Our preliminary experiments confirm that imputing the small number of missing evaluations does not significantly impact our analysis.

In what follows, we describe the datasets, ML models, quality metrics, and tuning strategies we use in our experiments. Then, we outline the different aggregation strategies. Appendix 7.2 provides further details about datasets and aggregation strategies.

### 4.1 Encoders

We used the category_encoders[1] implementations of Bin, CB, Count, Ord, OH, Sum, and WoE. We sourced MH from the authors' implementation [4, 5].[2] We implemented DMT, GLMM, ME, MT, PBMT, CVMT, BUMT, CVGLMM, and BUGLMM. We also added a baseline encoder, Drop, which encodes every level with 1. For DMT, we experimented with the number of bins: $\{2, 5, 10\}$, for ME, with the regularization strength: $\{0.1, 1, 10\}$, for PBMT, with the minimum frequency: $\{0.001, 0.01, 0.1\}$, and for cross-validated encoders, such as CVMT, with the number of folds: $\{2, 5, 10\}$. We display hyperparameter values with subscripts, e.g., $CV_2MT$.

### 4.2 Datasets

We used binary classification datasets. This allows us to conduct in-depth analysis using the same ML models and quality metrics. Additionally, certain supervised encoders, e.g., WoE, are specifically designed for binary classification tasks. We chose 50 datasets with categorical attributes from OpenML [42], including the suitable ones from the related work.

### 4.3 ML models

We experimented with diverse ML models that process data in different ways: decision trees (DT) and boosted trees (LGBM) exploit orderings, support vector machines (SVM) use kernels, k-nearest neighbors (k-NN) relies on distances, and logistic regression (LogReg) is a "pseudo-linear" model. The LGBM implementation we used is from the LightGBM module,[3] while the other models' implementations are from scikit-learn. Table 2 compares our model choices with related work. We

---

[1] https://contrib.scikit-learn.org/category_encoders/
[2] https://dirty-cat.github.io/stable/
[3] https://lightgbm.readthedocs.io/en/v3.3.5/

Table 2: ML models used in related studies.

| | | Ours | [28] | [4] | [8] | [5] | [44] | [41] |
|---|---|---|---|---|---|---|---|---|
| | Tree ensembles | ✓ | ✓ | ✓ | ✓ | ✓ | | ✓ |
| | Linear | ✓ | ✓ | | ✓ | | | ✓ |
| | SVM | ✓ | ✓ | | | ✓ | | ✓ |
| Model family | k-NN | ✓ | ✓ | | | | | |
| | DT | ✓ | ✓ | | ✓ | | ✓ | |
| | Neural | | | | ✓ | | | ✓ |
| | Naïve Bayes | | | | | | | ✓ |

excluded neural models due to their inferior performance on tabular data [15] and the absence of a recommended architecture. We also did not use Naïve Bayes due to its lack of popularity.

## 4.4 Quality metrics and tuning strategies

We assessed an encoder's quality by evaluating an ML model trained on the encoded data. We use four quality metrics: balanced accuracy (BAcc), F1-score (F1), accuracy (Acc), and Area Under the ROC Curve (AUC). We compared three tuning strategies:

- *no tuning*;
- *model tuning*: the entire training set is pre-processed before tuning the model;
- *full tuning*: the entire pipeline is tuned on the training set, with each training fold of the nested cross-validation pre-processed independently.

We used Bayesian search from scikit-optimize[4] for full tuning, and for model tuning grid search from scikit-learn. Table 4 summarizes the tuning search space for different ML models. To mitigate excessive runtime, we chose not to tune certain ML models and limited the dataset selection to the smallest 30 for full tuning, as Table 3 illustrates.

Table 3: Factors for different tuning strategies.

| | Models | # Datasets |
|---|---|---|
| No tuning | DT, k-NN, LogReg, SVM, LGBM | 50 |
| Model tuning | DT, k-NN, LogRreg | 50 |
| Full tuning | DT, k-NN, LogReg, SVM | 30 |

Table 4: Tuning search space.

| | Hyperparameter | Interval | Grid |
|---|---|---|---|
| DT | max_depth | $[2, \ldots, 5]$ | $\{2, 5, None\}$ |
| k-NN | n_neighbors | $[2, \ldots, 10]$ | $\{2, 5, 10\}$ |
| LogReg | C | $[0.2, 5]$ | $\{0, 1, 10\}$ |
| SVM | C | $[0.1, 2]$ | |
| | gamma | $[0.1, 100]$ | |

## 4.5 Aggregating into a consensus ranking

A common practice for summarizing and interpreting the results of benchmark experiments is to aggregate them into a *consensus ranking* of *alternatives* (encoders in our case) [10, 27, 15]. To obtain a dataset-independent ranking of encoders, we aggregate the results across different datasets while keeping all other factors fixed. We now present well-known aggregation strategies used in benchmarks.

**Heuristics** rank alternatives based on an aggregate score. Common aggregation heuristics include mean rank (R-M) [5], median rank (R-Md), mean quality (Q-M), median quality (Q-Md), rescaled

---

[4] https://scikit-optimize.github.io/stable/

mean quality [36, 15] (Q-RM), the number of times the alternative was ranked the best (R-B) or the worst (R-W) [41], the number of times the alternative's quality is better than the best quality multiplied by a threshold $\theta \leq 1$ (Q-Th$_\theta$).

**Friedman-Nemenyi tests** [10] (R-Nem$_{\text{p-value}}$). First, one ranks alternatives separately for each dataset and then applies a Friedman test to reject the hypothesis that all encoders have the same average rank. If the hypothesis is rejected, pairwise Nemenyi post-hoc tests are conducted to compare pairs of alternatives. Finally, one uses the results of these post-hoc tests to construct the consensus ranking. This aggregation strategy requires the user to choose a p-value.

**Kemeny-Young aggregation** [21, 47] (R-Kem) first ranks alternatives separately for each dataset. Then, it determines the consensus ranking that minimizes the sum of distances to the datasets' rankings. We adopt the approach described in [45], with a distance measure that accomodates ties and missing values in the rankings. We then formulate the optimization problem as a mixed integer linear problem and solve it using a GUROBI solver with academic license.[5] Kemeny-Young aggregation is much slower than the other aggregation strategies, taking minutes for each aggregation.

# 5 Results

This section summarizes the main results of our study. Appendix 7.3 further discusses the missing evaluations, run time, replicability, the ranks of the encoders and studies the effect of tuning on pipeline quality.

## 5.1 Sensitivity analysis

The relative performance of encoders, i.e., the ranking, can depend on the pick of ML model, quality metric, and tuning strategy. More, the choice of an aggregation strategy impacts the consensus ranking. To quantify the influence of these choices, we calculate the similarity between rankings using the Jaccard index $J$ for the sets of best encoders and the Spearman correlation coefficient $\rho$. Intuitively, $J$ measures if two experiments with different factor combinations agree on the best encoders, while $\rho$ takes the entire ranking into account. For both measures, values close to 1 indicate high agreement and low sensitivity. Conversely, values near 0 (or, for $\rho$, negative) suggest low consistency and high sensitivity.

### 5.1.1 Sensitivity to experimental factors

We evaluate the sensitivity of encoder rankings on individual datasets with respect to an experimental factor (ML model, quality metric, or tuning strategy) by varying the factor of interest and keeping the other factors fixed, then calculating the similarity between pairs of rankings. After that, we average the result across all combinations of the other factors. Figures 1a, 1b, and 1c show the resulting values, with Spearman's $\rho$ in the upper triangle and Jaccard index $J$ in the lower triangle. For example, Spearman's $\rho$ between encoder rankings for DT and SVM, averaged across all datasets, tuning strategies, and quality metrics, is 0.3.

Our findings highlight the high sensitivity of results to experimental factors, for both the full rankings and the best encoders. They also explain why results from other studies are so inconsistent, as choosing different values for any factor will lead to different results.

### 5.1.2 Sensitivity to aggregation strategy

To evaluate the impact of the aggregation strategy on the consensus ranking, we apply the same procedure as above to consensus rankings instead of rankings on individual datasets. Figure 1a presents the results with the notation from Section 4.5. For example, Spearman's $\rho$ between consensus rankings obtained with Q-M and Q-Md averaged across all ML models, tuning strategies, and quality metrics is 0.8.

While some aggregation strategies show strong similarities, different strategies yield very different consensus rankings in general. This is particularly evident for Jaccard index $J$, indicating the high sensitivity of the best encoders to the rank aggregation strategy.

---

[5]`https://www.gurobi.com/solutions/gurobi-optimizer/`

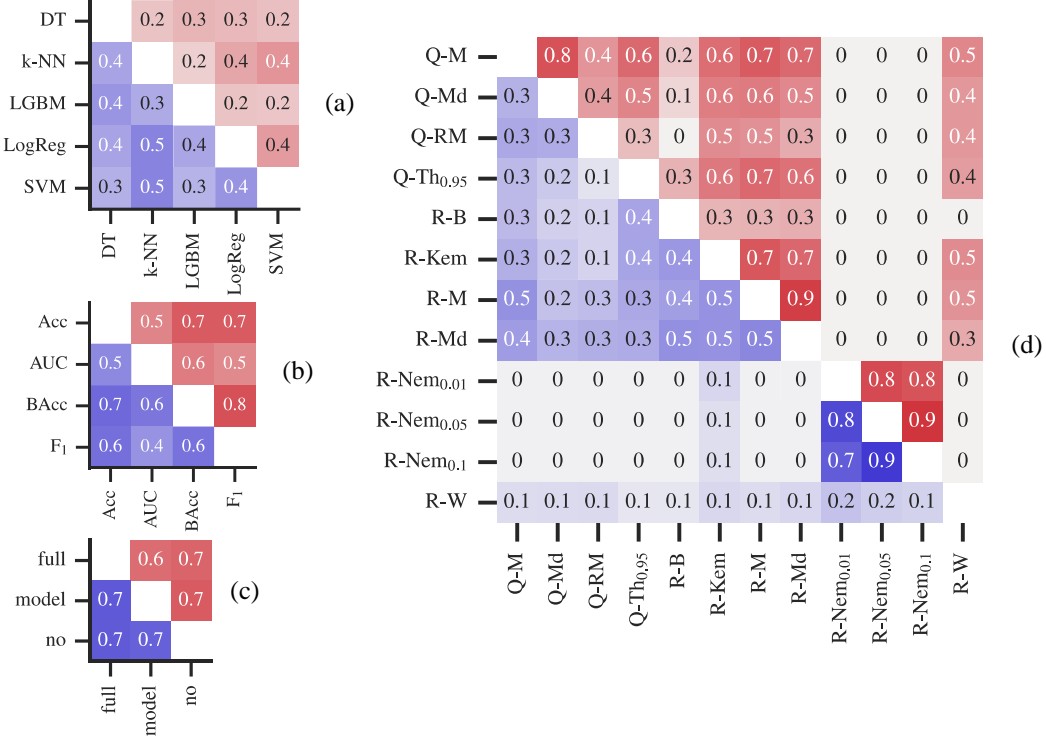

Figure 1: Sensitivity as the average similarity between rankings, measured with $\rho$ (upper triangle) and $J$ (lower triangle), computed between individual rankings for varying: (a) ML model, (b) quality metric, (c) tuning strategy, and between consensus rankings for varying (d) aggregation strategy.

## 5.2 Replicability

Replicability is defined as the property of a benchmark to produce consistent results from different data [26]. This definition does not, however, provide a quantifiable notion of replicability. To overcome this, we made the following modeling decisions. First, we fix a factor combination: ML model, quality metric, tuning strategy, and aggregation strategy. We excluded the R-Nem and R-Kem aggregation strategies due to their slower run time. Second, we model the result of a benchmark on a dataset sample $S$ with the consensus ranking aggregated across $S$. Third, we quantify replicability as the similarity between consensus rankings averaged over all factor combinations and 100 pairs of equal-sized disjoint sets of datasets. As discussed in Section 5.1, we measure the similarity with $\rho$ and $J$ to capture the similarity between both the rankings and the best encoders. We refer to them as $\rho$-replicability and $J$-replicability, respectively.

Figure 2 shows the outcome for different tuning strategies, conditional on the ML model and the size of the dataset samples. We have studied additional factors in Appendix 7.3.3. The shaded areas represent a bootstrapped 95% confidence interval. Our findings show an upward trend of $\rho$-replicability as the size of the dataset samples increases. This observation confirms that, in general, considering a larger number of datasets yields more reliable experimental outcomes. It is, however, important to note that this pattern does not always hold for $J$-replicability. This suggests that, for some models, the best encoders might vary significantly even with a relatively large number of datasets. To conclude, the replicability of our results strongly depends on the ML model, with logistic regression exhibiting the highest replicability and decision trees the lowest.

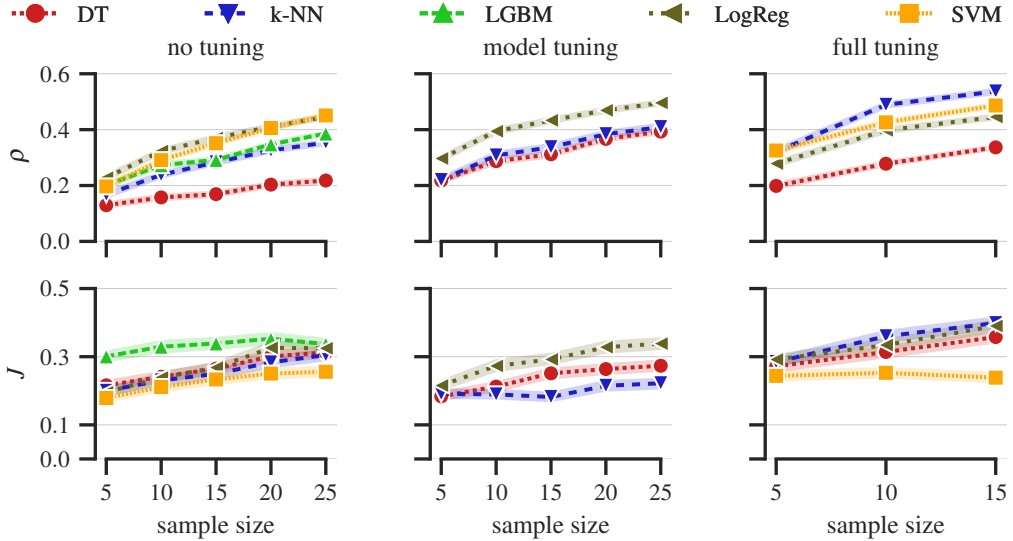

Figure 2: Replicability as the average similarity of consensus rankings from disjoint subsets of datasets.

## 5.3 Comparing encoders

Based on the outcome of Section 5.2, we now examine the ranks of encoders limited to decision trees, logistic regression, and all ML models.

Figure 3a shows the rank of encoders from the experiments with decision trees across all datasets, quality metrics, and tuning strategies. One-Hot is the best-performing encoder; however, Nemenyi tests at a significance level of 0.05 fail to reject that the average rank of One-Hot is the same as that of the other encoders.

Figure 3b features the encoder ranks for logistic regression, where four encoders, namely One-Hot, Sum, Binary, and Weight of Evidence, consistently achieve higher ranks compared to the others. Nemenyi tests confirm that this difference in ranks is significant. These results are in line with the ones from Section 5.2, which indicate low replicability of the results for decision trees and higher replicability for logistic regression.

Figure 3c presents the ranks of encoders across all datasets, ML models, quality metrics, and tuning strategies. Similarly to logistic regression, One-Hot, Sum, Binary, and Weight of Evidence consistently achieve significantly higher average ranks compared to the other encoders, again confirmed by Nemenyi tests. We recommend these four encoders as the preferred choices in practical applications. This conclusion contradicts other studies reporting a suboptimal performance of One-Hot [5, 28].

Our findings also reveal that Drop performs significantly worse than all other encoders, i.e., encoding categorical attributes generally yields better results than dropping them.

## 5.4 Comparing to related work

In this section, we compare our results with the findings of other studies. To do so, we select subsets of our results that mimic the experimental settings in related work. In [28], $CV_5GLMM$ outperformed every competitor for boosted trees and k-NN, while GLMM was recommended for SVMs. However, in our experiments, Sum outperformed GLMM for SVMs, One-Hot did better than $CV_5GLMM$ for boosted trees, and $CV_{10}GLMM$ was better than $CV_5GLMM$ for k-NN. Next, while in [5] similarity encoders are better than One-Hot for boosted trees, subsequent research reported no significant difference between Min-hash and One-Hot on medium-sized tabular datasets [4]. Our findings are in line with this latter result, as we could not find a performance difference between the two encoders with a t-test with a significance level of 0.05. In [41], Sum is reported as the best encoder on the Adult dataset for boosted trees, while a Data-constraining encoder is reported as the worst. With the same setting, we did not find a significant performance difference for any encoder except for Drop, which

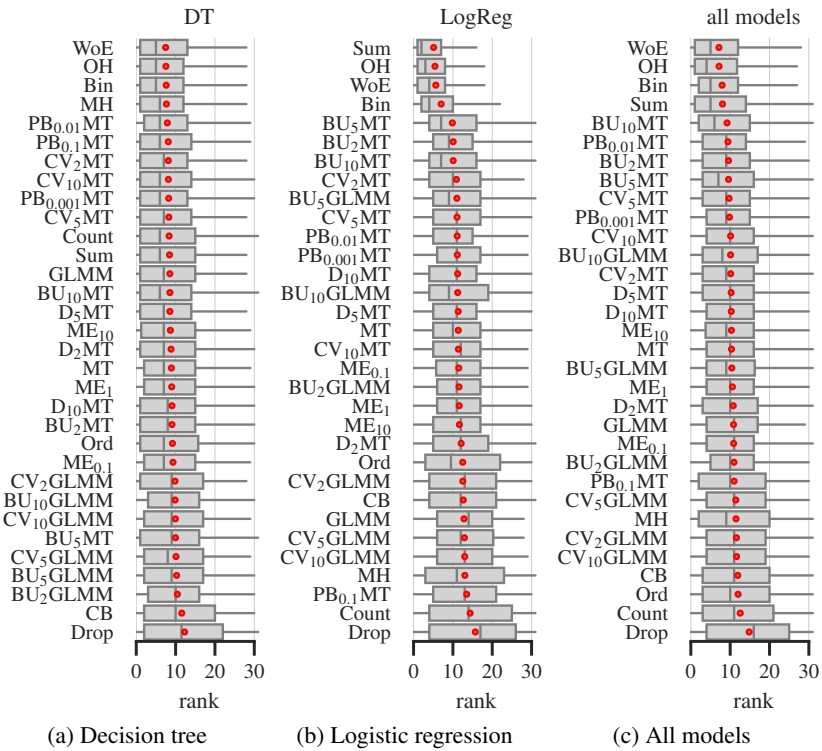

Figure 3: Ranks of encoders.

performed the worst. On the Bank marketing dataset, [8] showed that One-Hot and Mean-Target outperformed Binary with logistic regression. In our experiments, Binary was slightly worse than One-Hot and Mean-Target. In [44], Dummy, an identifier encoder similar to One-Hot, was better than Mean-Target on the Tic-tac-toe dataset with boosted trees. We, instead, did not observe any significant difference between One-Hot and Mean-Target for these factors.

## 6 Limitations and conclusions

**Limitations.** First, we treated encoders as part of the pre-processing, but certain encoders can be an integral component of specific ML models. For instance, CatBoost is derived from the homonymous boosted trees algorithm, which re-encodes the data multiple times during training. Second, we applied a single encoder to all categorical attributes. Using different encoders based on the cardinality of the attribute may sometimes yield favorable results [28, 4]. However, the selection of the optimal encoder for each attribute requires either domain knowledge of the attribute or purpose-built tools, which falls outside the scope of our benchmark and is therefore left as future work. We also did not include neural networks, due to the absence of a recommended architecture and reported interior performance to tree-based models on tabular data [15].

**Conclusions.**

In this study, we conducted an extensive evaluation of encoder performance across various experimental factors, including ML models, quality metrics, and tuning strategies. Our results demonstrate a high sensitivity of encoder rankings to these factors, both for the full rankings and the best-performing encoders. This sensitivity explains the inconsistent results among related studies, as different choices in any of these factors can lead to different outcomes. We also assessed the impact of aggregation strategies on consensus rankings, revealing significant variations in rankings depending on the chosen strategy. This emphasizes the importance of carefully considering the aggregation method when post-processing and interpreting results. Regarding replicability, we defined and quantified it using $\rho$-replicability and $J$-replicability. Our findings indicate that replicability is influenced by factors such as the ML model, with logistic regression exhibiting the highest replicability and decision trees

the lowest. Additionally, larger dataset samples tend to yield more reliable experimental outcomes, although this trend does not always hold for $J$-replicability. Based on our results, we recommend specific encoders for practical applications. For decision trees, Weight of Evidence performed the best, although statistical tests did not show a significant difference from other encoders. For logistic regression, Sum, One-Hot, Binary, and Weight of Evidence consistently achieved higher ranks, with statistically significant differences from other encoders. These findings contradict previous studies, highlighting the importance of considering a broad range of experimental factors. Finally, our comparative analysis with related work revealed discrepancies in encoder performance, suggesting that the breadth of our study may contribute to these differences. This emphasizes the need for caution when interpreting results from studies with more limited experimental settings. Overall, our study provides valuable insights into the sensitivity of encoder performance to experimental factors, as well as recommendations for practical encoder selection across different scenarios.

## Acknowledgments

We thank Dmitriy Simakov for valuable discussions and Natalia Arzamasova for her algorithm and implementation of the PreBinnedEncoder. This work was supported in part by the German Research Foundation (Deutsche Forschungsgemeinschaft), project *Charakterisierung, Modellierung und Homogenisierung von Vernetzungswerken mit Hilfe interpretierbarer Datenanalysemethoden*, and by the State of Baden-Württemberg, project *Algorithm Engineering für die Scalability Challenge*.

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

# 7 Appendix

## 7.1 Encoders

This section presents a reproducible description of the encoders discussed in Section 3, following the structure outlined below. We discuss identifier, frequency-based, contrast, and simple target encoders together in Appendix 7.1.4, as all of these encoders can be explicitly represented as functions. Similarity, binning, smoothing, and data-constraining encoders have dedicated sections. Table 5 contains the notation used in this section.

Table 5: Notation for section 7.1.

| Symbol | Meaning |
|---|---|
| $\mathbb{N}_0$ | natural numbers including 0 |
| $(x)_2$ | base-2 representation of $x \in \mathbb{N}_0$ |
| $X^{n \times d}$ | set of matrices with entries in $X$, $n$ rows and $d$ columns |
| $\mathbb{1}$ | indicator function |
| $D$ | binary classification dataset |
| $n$ | number of rows of $D$ |
| $\mathbf{y} \in \{0, 1\}^n$ | target attribute of $D$ |
| $\Omega_A = \{\omega_l\}_{l=1}^L$ | categorical domain (strings) |
| $\mathbf{A} \in \Omega_A^n$ | categorical attribute of $D$ to be encoded |
| $l_i \in \{1, \ldots, L\}$ | such that $\mathbf{A}_i = \omega_{l_i}$ |
| $E : \mathbf{A} \mapsto \mathbf{M} \in \mathbb{R}^{n \times d}$ | encoder |
| $\mathbf{M} \in \mathbb{R}^{n \times d}$ | encoding of $\mathbf{A}$, compact notation |
| $E(\mathbf{A}) \in \mathbb{R}^{n \times d}$ | encoding of $\mathbf{A}$ with explicit encoder |
| $E(\Omega_A)$ | unique values of rows of $E(\mathbf{A})$ |
| $d = d(E, \mathbf{A})$ | number of columns of $\mathbf{M}$ |
| $\mathbf{M}_i$ | $i$-th row of $\mathbf{M}$ if $d > 1$, $i$-th component of $\mathbf{M}$ if $d = 1$ |
| $l \in \{1, \ldots, L\}$ | index of levels |
| $i, h \in \{1, \ldots, n\}$ | row indices of $\mathbf{M}$, $\mathbf{A}$, or $\mathbf{y}$ |
| $j \in \{1, \ldots, d\}$ | column index of $\mathbf{M}$ |

### 7.1.1 Similarity encoders [5, 4]

**Min-Hash** treats $\omega \in \Omega_A$ as a string, splits it into its set of character-level n-grams (substrings of n-consecutive characters), uses a hash function to encode each n-gram into an integer, and finally encodes $\omega$ with the minimum value of the hash function on the set of n-grams. The process is repeated for $d$ hash functions, yielding $M \in \mathbb{R}^{n \times d}$. The default value of $d$ is 30, the authors report good performance with 300 [6].

### 7.1.2 Binning encoders

**Pre-Binned Mean-Target** partitions $\Omega_A$ into $B$ buckets $\{P_b\}_{b=1}^B$ to solve the optimization problem

$$\text{Maximize } B$$

$$\text{subject to } \frac{1}{n} \sum_{\omega \in P_b} \sum_{i=1}^n \mathbb{1}\left(\mathbf{A}_i = \omega\right) \geq \vartheta \qquad \forall b \leq B$$

where $\vartheta \in [0, 1]$ is a user-defined threshold. Each bucket is then treated as a new level and encoded with Mean-Target, yielding an encoding $M \in \mathbb{R}^n$.

**Discretized Mean-Target** partitions $MT(\Omega_A)$ into intervals $\{I_1, \ldots, I_B\}$ of equal length. Letting $I(l)$ be the interval that contains $MT(\omega_l)$ (that is, the average target associated to $\omega_l$), the encoding is $\mathbf{M} \in \mathbb{R}^n : \mathbf{M}_i = \inf I(l_i)$. We experimented with $B = 2, 5, 10$.

---

[6]`https://dirty-cat.github.io/stable/generated/dirty_cat.MinHashEncoder.html`

Table 6: Identifier encoders

| | $E(\Omega_A)$ | $E(\mathbf{A})$ |
|---|---|---|
| Binary [8] | $\{0,1\}^{[\log_2(L)]+1}$ | $\mathbf{M}_i = (l_i)_2$ |
| Dummy [28, 44] | $\{0,1\}^{L-1}$ | $\mathbf{M}_{ij} = \begin{cases} \mathbb{1}(\mathbf{A}_i = \omega_j) & j \neq L \\ 0 & j = L \end{cases}$ |
| One-Hot [28, 4, 8, 44, 41] | $\{0,1\}^L$ | $\mathbf{M}_{ij} = \mathbb{1}(\mathbf{A}_i = \omega_j)$ |
| Ordinal [28, 44, 41] | $\mathbb{N}_0$ | $\mathbf{M}_i = l_i$ |

Table 7: Frequency-based encoders

| | | |
|---|---|---|
| Count [36] | $\mathbb{N}_0$ | $\mathbf{M}_i = \sum_j \mathbb{1}(\mathbf{A}_j = \omega_{l_i})$ |
| Frequency [28] | $\mathbb{R}$ | $\mathbf{M}_i = \frac{1}{n} \sum_j \mathbb{1}(\mathbf{A}_j = \omega_{l_i})$ |

Table 8: Contrast encoders — without intercept

| | | |
|---|---|---|
| Sum [41] | $\mathbb{R}^{L-1}$ | $\mathbf{M}_{ij} = \begin{cases} \mathbb{1}(\mathbf{A}_i = j) & j \neq L \\ -1 & j = L \end{cases}$ |
| Backward difference [41] | $\mathbb{R}^{L-1}$ | $\mathbf{M}_{ij} = \begin{cases} -\frac{L-i}{L} & i \leq j \\ \frac{1}{L} & i > j \end{cases}$ |
| Helmert [41] | $\mathbb{R}^{L-1}$ | $\mathbf{M}_{ij} = \begin{cases} -\frac{1}{j+1} & i \leq j \\ \frac{j}{j+1} & i = j+1 \\ 0 & i \geq j+2 \end{cases}$ |

Table 9: Simple target encoders

| | | |
|---|---|---|
| Mean-Target [28, 8] | $\mathbb{R}$ | $\mathbf{M}_i = \sum\limits_{h=1}^{n} y_h \mathbb{1}(\mathbf{A}_h = \omega_{l_i})$ |
| Weight of Evidence [39] [28] | $\mathbb{N}_0$ | $\mathbf{M}_i = \log\left(\frac{MT(\mathbf{A})}{1-MT(\mathbf{A})}\right)$ |

### 7.1.3 Smoothing target encoders

**Mean-Estimate** [41]. Let $n_l = \sum_{i=1}^{n} \mathbb{1}(A_i = \omega_l)$ be the number of occurrences of $\omega_l$ in $\mathbf{A}$.

$$\mathbf{M}_i = \frac{n_{l_i} MT(\omega_{l_i}) + \frac{w}{n} \sum\limits_{i=1}^{n} y_i}{w + n_{l_i}}$$

where $w$ is a user-defined weight. Common choices are $1, 10$.

**GLMM** [28] fits, for every $\omega_l \in \Omega_A$, a random intercept model

$$y_i = \beta_{l_i} + u_{l_i} + \varepsilon_i$$

where $u_l \sim N(0, \tau^2)$ and $\varepsilon_i \sim N(0, \sigma^2)$. The encoding is $\mathbf{M} \in \mathbb{R}^n : \mathbf{M}_i = \beta_{l_i}$.

### 7.1.4 Identifier, frequency-based, contrast, simple target encoders

The descriptions are divided as follows: Table 6 is for identifier encoders, Table 7 is for frequency-based encoders, Table 8 is for contrast encoders, and Table 9 is for simple target encoders.

### 7.1.5 Data-constraining encoders

**CatBoost** [41] uses a permutation $\pi$ of $\{1, \ldots, n\}$ and encodes with $\mathbf{M} \in \mathbb{R}^n$ such that

$$\mathbf{M}_{\pi(i)} = \sum_{h \leq \pi(i)} y_h \mathbb{1}\left(A_h = \omega_{l_{\pi(i)}}\right)$$

Table 10: Notation for section 7.2.2.

| Symbol | Meaning |
|---|---|
| $\perp$ | missing evaluation or rank |
| $\mathbb{1}$ | indicator function |
| $E_i$ | encoder as in table 5 |
| $\Phi_j : E_i \mapsto \mathbb{R} \cup \{\perp\}$ | average cross-validated quality on the $j$-th dataset, all other factors fixed |
| $\Phi_j^{\max} = \max_{i=1,\ldots,n} \{\Phi_j(E_i)\}$ | best quality on the $j$-th dataset, all other factors fixed |
| $\Phi_j^{\min} = \min_{i=1,\ldots,n} \{\Phi_j(E_i)\}$ | worst quality on the $j$-th dataset, all other factors fixed |
| $r_j : E \mapsto \mathbb{N}_0 \cup \{\perp\}$ | ranking obtained from $\Phi_j$ |
| $\mathbf{R}^j = (\mathbb{1}\,(r_j(E_i) \leq r_j(E_h)))_{i,h=1}^n \in \{0,1\}^{n \times n}$ | adjacency matrix of $r_j$ |
| $c : E \mapsto \mathbb{N}_0 \cup \{\perp\}$ | consensus ranking |
| $\mathbf{C} \in \{0,1\}^{n \times n}$ | adjacency matrix of $c$ |
| $i, h, k \in \{1, \ldots, n\}$ | index of encoders |
| $j \in \{1, \ldots, m\}$ | index of objects to be aggregated |

**Cross-Validated MT** [28] randomly partitions $\{1, \ldots, n\}$ in $k$ folds of equal size. Let $D_{a_i}$ be the fold that contains $i$. Then, every fold is encoded with Mean-Target trained on the other $k-1$ folds:

$$\mathbf{M}_i = \sum_{h=1}^{n} \mathbb{1}\,(h \notin D_{a_i})\,\mathbb{1}\,(\mathbf{A}_h = \omega_{l_i})\,y_h$$

Common values for $k$ are $2, 5, 10$.

**Cross-Validated GLMM** [28] works in a similar fashion as CVMT: it encodes each fold with GLMM trained on the other $k-1$ folds.

**BlowUp Cross-Validated MT** randomly partitions $\{1, \ldots, n\}$ in $k$ folds $D_1, \ldots, D_k$ of roughly equal size. Then, it encodes with $\mathbf{M} \in \mathbb{R}^{n \times k}$ so that the $j$-th column is $\mathbf{A}$ encoded with Mean-Target trained on the $j$-th fold, yielding

$$\mathbf{M}_{ij} = \sum_{h=1}^{n} \mathbb{1}\,(h \in D_j)\,\mathbb{1}\,(\mathbf{A}_h = \omega_{l_i})\,y_h$$

We experimented with $k = 2, 5, 10$.

**Blowup Cross-Validated GLMM** is analogous to BUMT, but the $j$-th column of its encoding $\mathbf{M}$ is encode with $\mathbf{M} \in \mathbb{R}^{n \times k}$ so that the $j$-th column of $\mathbf{M}$ is $\mathbf{A}$ encoded with GLMM trained on the $j$-th fold.

## 7.2 Experimental design

This section provides additional details about the datasets and aggregation strategies we discussed in Section 4. The notation we use in this section is summarized in Table 10.

### 7.2.1 Datasets

Table 11 lists the datasets used in our experiments. The columns are as follows: ID is the OpenML identifier; $n$ is the number of rows; $d$ is the number of attributes; $d_{cat}$ is the number of categorical attributes; $\max|\Omega_A|$ is the maximum categorical attribute cardinality; the "ft" flag denotes datasets used for full-tuning (cf. Section 4.4)

Table 11: Datasets used in the study.

| Name | Ref. | ID | $n$ | $d$ | $d_{cat}$ | $\max\lvert\Omega_A\rvert$ | ft |
|---|---|---|---|---|---|---|---|
| ada_prior | | 1037 | 4562 | 14 | 7 | 40 | ✓ |
| adult | | 1590 | 48842 | 14 | 7 | 42 | ✓ |
| airlines | | 1169 | 539383 | 7 | 4 | 293 | |
| amazon_employee_access | | 4135 | 32769 | 9 | 9 | 7518 | |
| Agrawal1 | | 1235 | 1000000 | 9 | 3 | 20 | |
| Australian | [33] | 40981 | 690 | 14 | 4 | 14 | ✓ |
| bank-marketing | [24] | 1461 | 45211 | 16 | 6 | 12 | |
| blogger | [13] | 1463 | 100 | 5 | 3 | 5 | ✓ |
| Census-Income-KDD | | 42750 | 199523 | 41 | 27 | 51 | |
| credit-approval | [32] | 29 | 690 | 15 | 6 | 15 | ✓ |
| credit-g | [17] | 31 | 1000 | 20 | 11 | 10 | ✓ |
| cylinder-bands | [12] | 6332 | 540 | 37 | 17 | 71 | ✓ |
| dresses-sales | [40] | 23381 | 500 | 12 | 11 | 25 | ✓ |
| heart-h | [19] | 51 | 294 | 13 | 6 | 4 | |
| ibm-employee-attrition | | 43896 | 1470 | 34 | 5 | 9 | ✓ |
| ibm-employee-performance | | 43897 | 1470 | 33 | 5 | 9 | ✓ |
| irish | | 451 | 500 | 5 | 2 | 11 | ✓ |
| jungle_chess_2pcs..._elephant | [35] | 40999 | 2351 | 46 | 2 | 3 | ✓ |
| jungle_chess_2pcs..._lion | [35] | 41007 | 2352 | 46 | 2 | 3 | ✓ |
| jungle_chess_2pcs..._rat | [35] | 41005 | 3660 | 46 | 2 | 3 | ✓ |
| kdd_internet_usage | | 981 | 10108 | 68 | 20 | 129 | ✓ |
| KDDCup09_appetency | | 1111 | 50000 | 230 | 33 | 15416 | |
| KDDCup09_churn | | 1112 | 50000 | 230 | 33 | 15416 | |
| KDDCup09_upselling | | 1114 | 50000 | 230 | 33 | 15416 | |
| KDD98 | | 42343 | 82318 | 477 | 107 | 18543 | |
| kr-vs-kp | [37] | 3 | 3196 | 36 | 1 | 3 | ✓ |
| kick | | 41162 | 72983 | 32 | 17 | 1063 | |
| law-school-admission-bianry | | 43890 | 20800 | 11 | 1 | 6 | |
| molecular-biology_promoters | [16] | 956 | 106 | 57 | 56 | 4 | ✓ |
| monks-problems-1 | [43] | 333 | 556 | 6 | 4 | 4 | ✓ |
| monks-problems-2 | [43] | 334 | 601 | 6 | 4 | 4 | ✓ |
| mv | | 881 | 40768 | 10 | 1 | 3 | ✓ |
| mushroom | [25] | 43922 | 8124 | 22 | 16 | 12 | ✓ |
| national-longitudinal-survey-binary | [6] | 43892 | 4908 | 16 | 4 | 29 | ✓ |
| nomao | [3] | 1486 | 34465 | 118 | 27 | 3 | |
| nursery | | 959 | 12960 | 8 | 7 | 5 | |
| open_payments | | 42738 | 73558 | 5 | 4 | 4374 | |
| porto-seguro | | 41224 | 595212 | 57 | 13 | 104 | |
| profb | | 470 | 672 | 9 | 3 | 28 | ✓ |
| sick | [34] | 38 | 3772 | 29 | 2 | 5 | ✓ |
| sf-police-incidents | | 42344 | 538638 | 6 | 5 | 21838 | |
| SpeedDating | | 40536 | 8378 | 120 | 58 | 260 | ✓ |
| students_scores | | 43098 | 1000 | 7 | 2 | 6 | ✓ |
| telco-customer-churn | | 42178 | 7043 | 19 | 11 | 6531 | |
| thoracic-surgery | [48] | 1506 | 470 | 16 | 3 | 7 | ✓ |
| tic-tac-toe | [1] | 50 | 958 | 9 | 9 | 3 | ✓ |
| Titanic | | 40945 | 1309 | 13 | 6 | 1307 | |
| vote | [6] | 56 | 435 | 16 | 16 | 3 | ✓ |
| wholesale-customers | | 1511 | 440 | 8 | 1 | 3 | ✓ |
| WMO-Hurricane-Survival-Dataset | | 43607 | 5021 | 22 | 21 | 4173 | |

### 7.2.2 Aggregation strategies

This section presents the mathematical formulations of the aggregation strategies. As Section 4.5 explains, the results are aggregated across datasets, while keeping the other factors — ML model, tuning strategy, and quality metric — fixed.

**Heuristics.** Heuristics aggregate by ranking encoders according to some score. *Increasing* heuristics assign the best rank to the encoder with the highest score, while the non-increasing ones assign the best rank to the encoders with the lowest score. Table 12 contains the respective formulas. Any missing evaluations ($\perp$) are ignored during the computation.

**Friedman-Nemenyi tests.** The Friedman test is used to rule out the null hypothesis that all encoders have, on average, the same rank. The Friedman statistic adjusted for ties [38, 18] is

$$T = \frac{(m-1)(S_t - C)}{S_r - C}$$

where $S_r = \sum_{i=1}^{n} \sum_{j=1}^{m} r_j(E_i)^2$, $S_t = \frac{1}{m} \sum_{i=1}^{n} \left( \sum_{j=1}^{m} r_j(E_i) \right)^2$, and $C = \frac{1}{4} mn(n+1)^2$.

Under the null hypothesis that all encoders have the same rank, $T$ is approximately distributed as an $F$-distribution with $n-1$ and $(m-1)(n-1)$ degrees of freedom.

If the Friedman hypothesis is rejected, one can compare all pairs of encoders with $n(n-1)/2$ Nemenyi post-hoc tests [10]. Nemenyi tests apply a correction to control the error from testing multiple hypotheses. Two encoders $E_1$ and $E_2$ are significatively different if

$$\frac{1}{m} \sum_{j=1}^{m} (r_j(E1) - r_j(E_2)) \geq q_\alpha \sqrt{\frac{n(n+1)}{6m}}$$

where $\frac{1}{\sqrt{2}} q_\alpha$ is the critical value based on a Studentized range statistic [10].

**Kemeny-Young aggregation.** [21, 47, 45, 23]

The consensus's adjacency matrix $\mathbf{C}$ is a solution to the mixed-integer linear problem

$$\text{Maximize} \sum_{i,h} \mathbf{S}_{ih}(2\mathbf{C}_{ih} - 1)$$
$$\text{subject to } \mathbf{C}_{ih} - \mathbf{C}_{kh} - \mathbf{C}_{ik} \geq -1 \qquad \forall i \neq h \neq k \neq i$$
$$\mathbf{C}_{ih} + \mathbf{C}_{hi} \geq 1 \qquad \forall i < h$$
$$\mathbf{C}_{ih} \in \{0, 1\} \qquad \forall i, h$$

where $\mathbf{S} = \left( \sum_j \frac{\mathbf{R}_{ih}^j}{n_j(n_j - 1)} \right)_{i,h=1}^{n}$ is a cost matrix and $n_j = \sum_i \mathbb{1}(r_j(E_i) \neq \perp)$ is the number of encoders with evaluation on dataset $j$. This formulation accounts for ties and missing ranks.

### 7.3 Results

This section complements Section 5.

### 7.3.1 Missing evaluations

We successfully completed 61812 runs out of 64000, one per combination of encoder, dataset, ML model, tuning strategy, and quality metric. The 2188 failed evaluations are equally distributed among the encoders, while tuning was the greatest influencing factor. Indeed, there were 4303 missing runs with no tuning, 1152 with model tuning, and 32 with full tuning. This is likely due to the bigger datasets used in no tuning and model tuning, cf. Section 4.4 and Table 11. The total runtime for successful evaluations was 108 days.

Table 12: Scores of heuristics.

| | **Score of $E$** | **Increasing** |
|---|---|---|
| Mean rank | $\frac{1}{m}\sum_j r_j(E)$ | |
| Median rank | $\mathrm{median}\left(\{r_j(E)\}_{j=1}^{m}\right)$ | |
| Rank best | $\sum_{j=1}^{m} \mathbb{1}\left(r_j(E)=1\right)$ | ✓ |
| Rank worst | $\sum_{j=1}^{m} \mathbb{1}\left(r_j(E) \neq \max_{i=1,\dots,m} r(E_j)\right)$ | |
| Mean quality | $\frac{1}{m}\sum_j \Phi_j(E)$ | ✓ |
| Median quality | $\mathrm{median}\left(\{\Phi_j(E)\}_{j=1}^{m}\right)$ | ✓ |
| Rescaled mean quality | $\frac{1}{m}\sum_{j=1}^{m} \frac{\Phi_j(E)-\Phi_j^{\min}}{\Phi_j^{\max}-\Phi_j^{\min}}$ | ✓ |
| $\vartheta$-best quality | $\sum_{j=1}^{m} \mathbb{1}\left(\Phi_j(E) \geq \vartheta \cdot \Phi_j^{\max}\right)$ | ✓ |

#### 7.3.2 Run time

We computed two scores for the runtimes of encoders. First is the time necessary to encode the dataset. The outcome, displayed in figure 4a, is that GLMM-based encoders are the slowest. This happens because the bottleneck of these encoders is the fitting of the random intercept model, a problem that we could only partially alleviate with our custom implementation. As expected, the model has no influence on the runtime.

Second, the time necessary to tune the model-encoder pipeline, each tuning step requiring encoding the dataset and then fitting a model. Figure 4b tells a similar story as for encoding, with GLMM-based encoders being the slowest. The other encoders, apart from Drop and Mean-Target, all show a similar runtime.

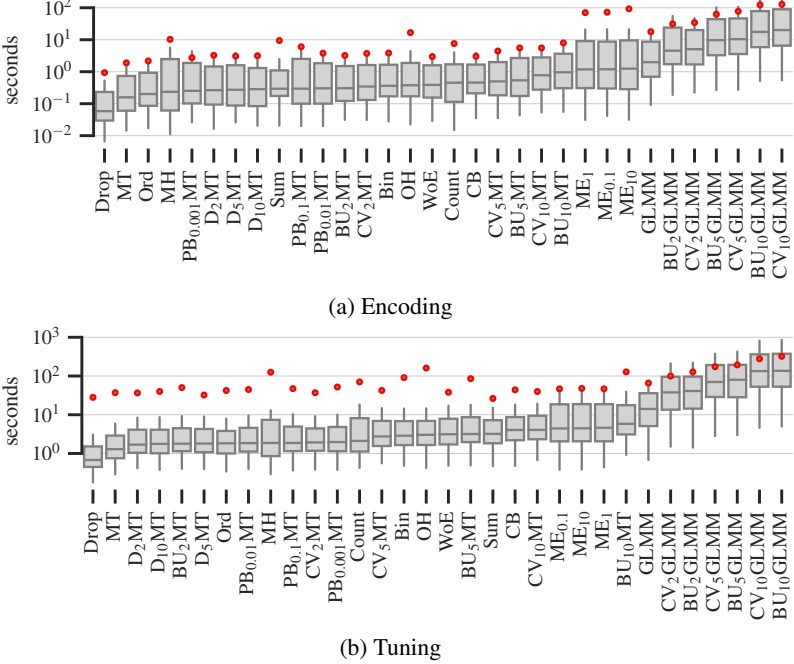

(a) Encoding

(b) Tuning

Figure 4: Runtime of encoders (a) and full tuning pipelines (b).

### 7.3.3 Replicability

This section extends the replicability analysis of Section 5.2, showing the behavior of different quality metrics in Figure 5a and aggregation strategies in Figure 5b.

The quality metrics behave similarly for $\rho$-replicability. The notable exception is the AUC in model tuning, which is significantly better than the other metrics. Regarding $J$-replicability, instead, accuracy is clearly the poorer choice. This hints that accuracy cannot discern the best encoder as well as the other metrics do and that it is more sensitive to the choice of dataset.

Among the aggregation strategies, rank best (R-B) shows higher replicability. A possible explanation is that R-B produces consensus rankings with many encoders tied as the best ones and few tiers in general,

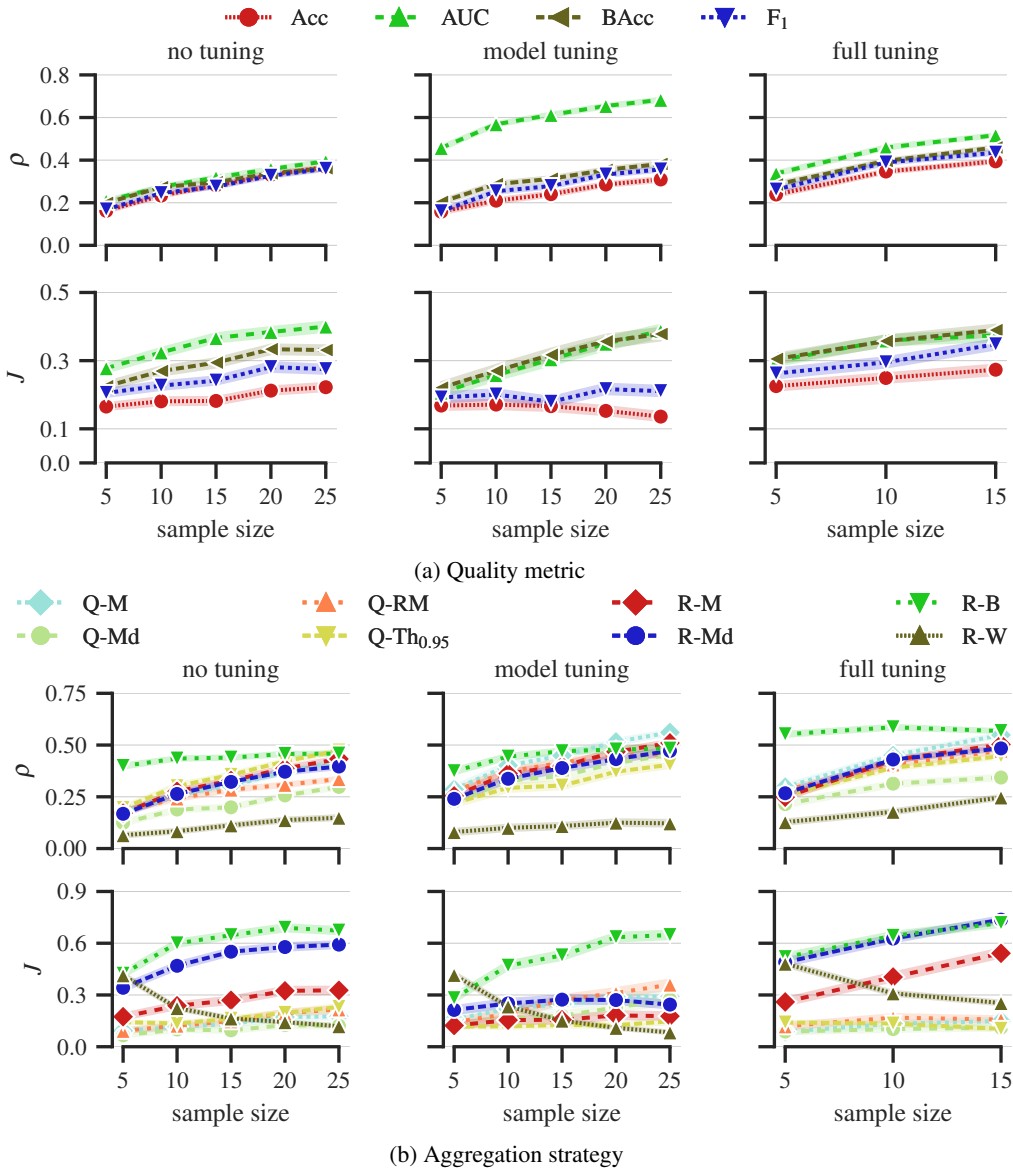

Figure 5: Average similarity of consensus rankings from disjoint subsets of datasets, conditional on (a) quality metric and (b) aggregation strategy.

### 7.3.4 Comparing encoders

This section expands on Section 5.3 and portrays in Figure 6 the distribution of ranks of encoders. The best encoders are evident for LogReg (Sum, OH, WoE, Bin) and k-NN (WoE), confirmed by Nemenyi tests at $0.05$ significance.

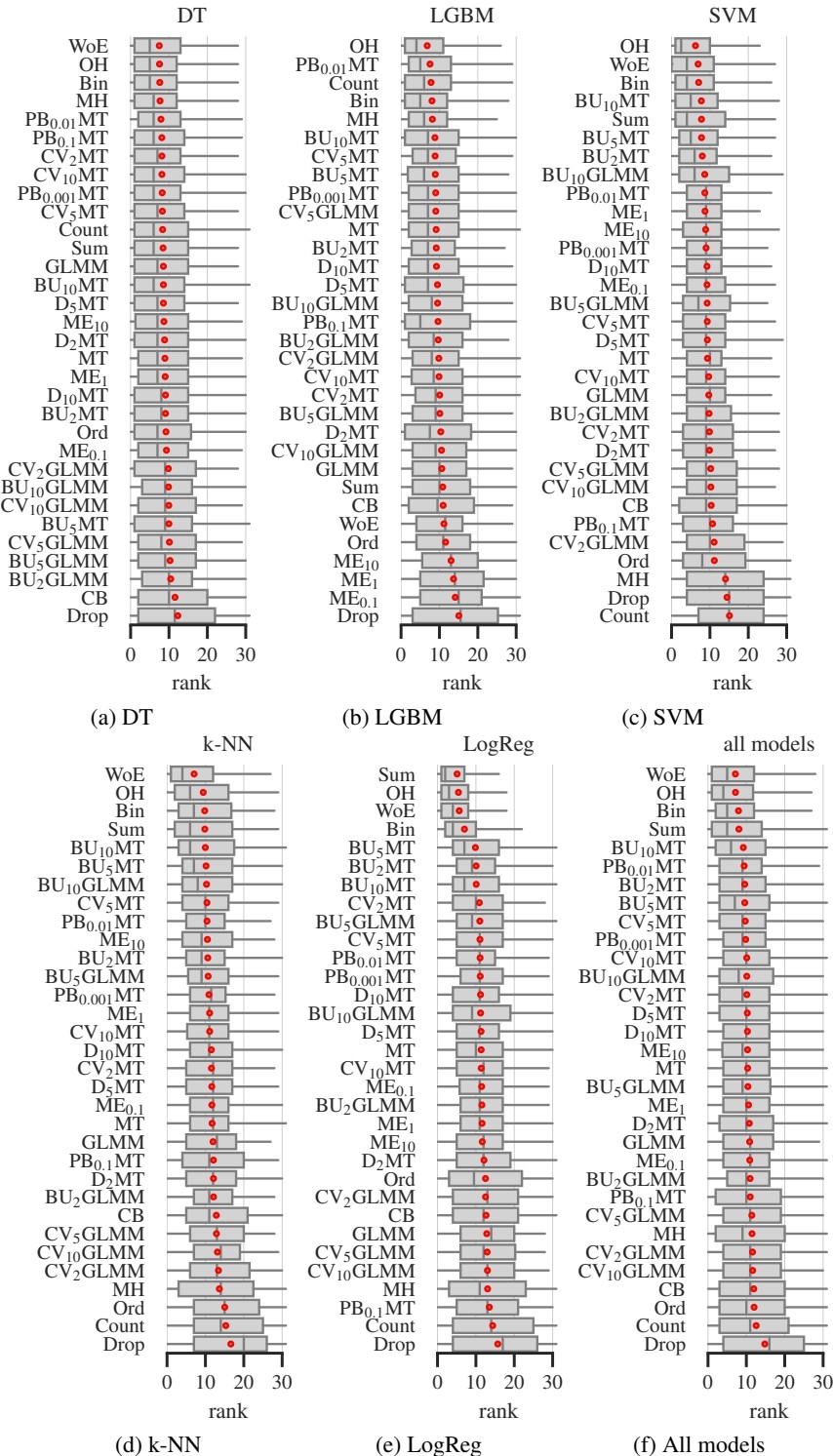

Figure 6: Ranks of encoders.

### 7.3.5 Effect of tuning

This section investigates whether tuning leads to improvements in pipeline performance. The tuning strategies are described in Section 4.4 For a pair of tuning strategies, we consider the factors they share and subtract the performance of the pipelines. Figure 7 shows that full tuning is, in general, advantageous over no tuning and slightly better than model tuning.

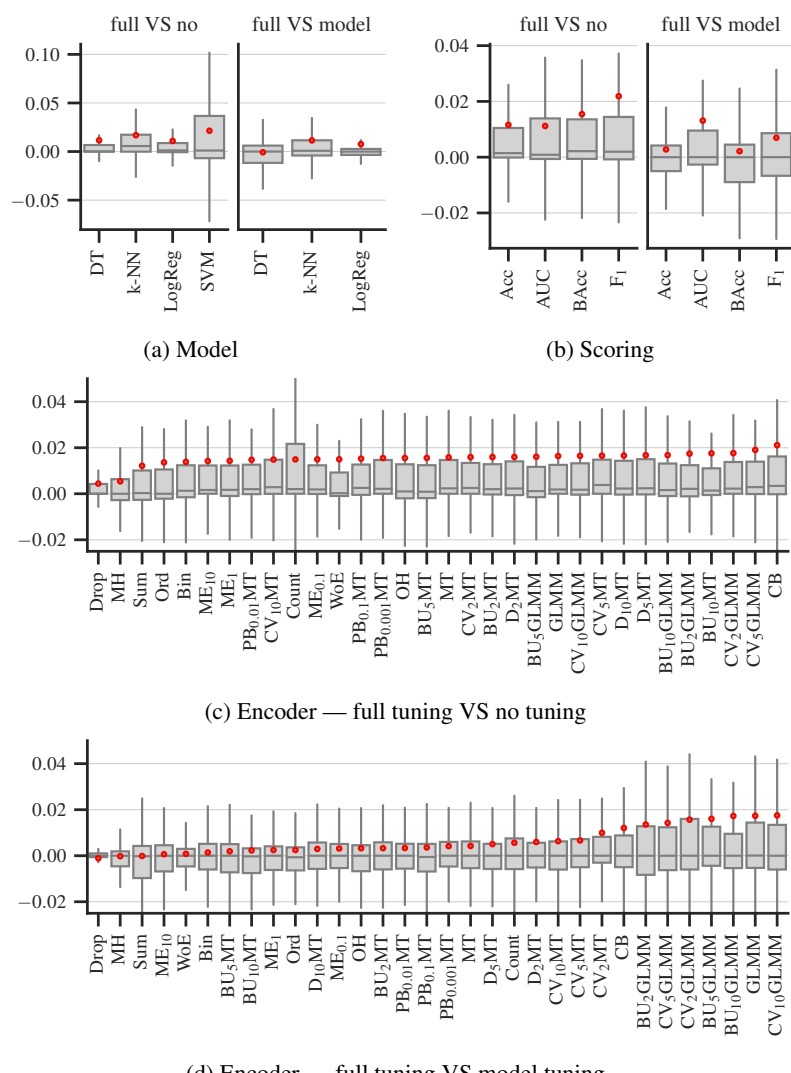

Figure 7: Performance gain of full tuning over no tuning and model tuning.

