# OpenReview forum: "A benchmark of categorical encoders for binary classification"
_NeurIPS.cc/2023/Track/Datasets_and_Benchmarks — NeurIPS 2023 Datasets and Benchmarks Poster_

### Official Review · Reviewer_QEyt · 2023-07-10
**Not anonymous**

**Rating:** 1
**Confidence:** 5
**Correctness:** N/A
**Clarity:** N/A

**Strengths:**

N/A

**Additional Feedback:**

N/A

**Documentation:**

N/A

**Opportunities For Improvement:**

N/A

**Relation To Prior Work:**

N/A

**Summary And Contributions:**

The paper is not anonymous.

---

### Official Review · Reviewer_N1YT · 2023-07-20
**Well-written paper, extensive evaluation, but maybe an issue with AUC?**

**Rating:** 7
**Confidence:** 4

**Strengths:**

The evaluation is extensive and the experimental setup is varied across many different axes. The paper not only reports on the final ranking of encoding strategies, but clearly shows the various ways the effect of certain experimental design decisions can affect the rankings. The figures are easy to parse, and the text is well-written. It benchmarks many different encoding strategies.

**Additional Feedback:**

The AUC issue needs to be addressed (or clarified, in case I misunderstand), but otherwise I think it is a strong paper.

**Clarity:**

Yes. Though it is not clear to me how to interpret the "grid" column of Table 4. Is it the discretized space of "interval" used in grid search? If so, why does logreg-c's interval upperbound at 5, but the grid has value 10?

**Correctness:**

From the docs it reads (https://github.com/DrCohomology/EncoderBenchmarking):
 - a Model implements the `fit`, `predict`, and `fit_predict` methods;
 - a quality metric is a function with signature `(y_true, y_pred) -> float`. Edit the corresponding parameters in src\config.py to add the objects to the benchmark.

And the source reads (https://github.com/DrCohomology/EncoderBenchmarking/blob/897fef7b9deb7b2053f9242896ca3caa1d3b8e09/src/main_no_tuning.py#L110):

`"cv_score": scoring(yte, model.predict(prepipe.transform(Xte))),`


One of the metrics in the paper is AUC, but above makes me think that predicted labels, not predicted probabilities, are used. For AUC probabilities should be used instead. It should affect results from those experiments, though I do not think it changes the overall findings.

**Documentation:**

The Github page seems to have enough documentation to run the experiments (not tested). Data is provided though could be more explicitly mentioned in the readme, it only mentions the subfolder it can be found in. Additionally, the result files should be accompanied by a description of what data they contain and what commands we used to obtain them.

**Ethics:**

No.

**Limitations:**

Yes.

**Opportunities For Improvement:**

Various software packages (e.g., scikit-learn) only have a footnote as reference, though these packages often also have appropriate papers to cite and explicitly ask you to do so (e.g., https://scikit-learn.org/stable/about.html#citing-scikit-learn). The footnotes to the documentation pages are fine, but should at least be accompanied by a citation where one is available.

Table 1 has a number of question marks where the tuning method is not described in the paper. I think it would be good to contact the authors of those respective papers (and/or inspect their code) to, as much as possible, complete the table.

Not all experiments ran to completion, about ~4% errored (L122-124). It is not clear to me which strategy is applied to account for the missing values, are they simply ignored? Imputed, how? This should be explicit and motivated.

Applying the same encoding scheme across all features in a dataset is unrealistic. This is something the authors explicitly mention as a limitation. I do not think the authors should address it in the review period, but I see it as the main opportunity for improvement in follow up work.

**Relation To Prior Work:**

Yes, but it would be good to mention related work on analysis of benchmarks. In particular, the work by Bouthillier et al. (2021) also breaks down variance contributions by different design decisions in the benchmark and model.


Bouthillier, Xavier, et al. "Accounting for variance in machine learning benchmarks." Proceedings of Machine Learning and Systems 3 (2021): 747-769.

**Summary And Contributions:**

The authors benchmark categorical encoding methods for binary classification, and measure the effect of various aspects in the experimental setup (such as tuning of the algorithms). They find that:

 - Benchmarks in this area need a high number of datasets (>25) to be replicable, which is generally more than papers proposing new techniques typically use.
 - Conclusions are still highly dependent on model, evaluation metric and aggregation method.

When comparing methods across all the different experimental setups, authors find that one-hot, sum, binary, and WoE encodings are the only ones to have a significantly better rank from other encoder (and this does not even hold true when working with DTs). This contradicts some earlier studies.

There might be a critical issue with the AUC evaluations (see: Correctness), though it _probably_ does not affect the overall results.

---

> ### Author Response · Authors · 2023-08-17
> **Response to reviewer N1YT**
>
> We thank the reviewer for the very insightful comments and feedback. In what follows, we provide answers to your comments and point to the revised sections of text, highlighting changes in blue.
>
> ***1. Package citation.***
>
> Thank you for bringing this to our attention. We have included the required citations for scikit-learn (L126) and dirty-cat (L141) in Section 4 of the revised manuscript. Additionally, we have ensured that the paper aligns with the citation policies of the other packages.
>
> ***2. Question marks in Table 1.***
>
> We have adhered to the recommendations and have successfully updated the table with appropriate replacements for three out of the four question marks. We are currently awaiting the authors' response for the remaining one.
>
> ***3. Handling incomplete experiments.***
>
> We only considered the successful evaluations to compute the rankings, disregarding missing evaluations. We did so (1) to avoid introducing unnecessary variability due to different imputation techniques, and (2) because there is no clearly superior imputation method. Due to the small number of missing evaluations, this choice should minimally impact our analysis and conclusions. To validate this, we conducted additional experiments assigning worst ranks to missing evaluations, observing only minor changes in results in sensitivity, replicability, and encoder comparison. For instance, changes in the cells of Figure~1d are, in absolute value, less than $0.05$ in $>80\%$ of the cells.
> These clarifications and motivations are now part of Section 4 of our revised manuscript (L131--135).
>
> ***4. Feature-based encoding.***
>
> We concur with this comment and recognize the various challenges associated with tackling this open problem. For instance, selecting the appropriate encoder scheme for individual categorical attributes often requires domain expertise and human judgment. To facilitate large-scale experiments, automating this procedure becomes imperative. Next, assessing the impact of each encoder presents a complex undertaking with debatable benefits. Our opinion is that a more useful result would involve devising a methodology for encoder selection based on meta-features. Consequently, the emphasis transitions from merely benchmarking encoders to introducing an AutoML approach. We expanded Section 6 to include these motivations (L275--277).
>
> ***5. AUC.***
>
> We highly appreciate your dedication to reviewing our code and identifying this inconsistency. You are correct, and our analysis employs predicted class, not probability. Hence, we report balanced accuracy rather than ROC AUC. We have updated the paper and the readme in the repository accordingly.
>
> We plan to incorporate ROC AUC in our study. Our ongoing experiments, though requiring a few more weeks to finalize, have already yielded partial results for decision tree, logistic regression, and non-GLMM-based encoders.  These preliminary findings confirm your expectations regarding the limited influence of these updates on overall conclusions. In particular, the rankings of encoders for logistic regression and decision tree are unchanged, especially for the top-performing encoders. These plots are now based on all four quality metrics: accuracy, balanced accuracy, F1, and ROC AUC.
> Upon experiment completion, we will update the paper accordingly, but we have (temporarily) included these results in the revised supplementary material, Appendix 7.3.4 (L506--514).
>
> ***6. Tuning search space (Table 4).***
>
> In Section 4.4, we used distinct tuning methods for our pipelines: an iterative Bayesian search on continuous intervals for full tuning, and a grid search on discrete values for model tuning. This decision was driven by our preference to retain all 50 datasets for model tuning and 30 smaller datasets for full tuning, owing to computational limitations. We acknowledge the slight inconsistency in the search ranges you observed and have addressed this by aligning search ranges in the new experiments mentioned earlier. Following the completion of the experiments, we will incorporate corresponding updates into the paper. As before, we anticipate little to no impact on our findings.
>
> ***7. Related work on analysis of benchmarks.***
>
> We have now included an additional paragraph in Section 2 of the revised manuscript (L60--66), providing an overview of related work in benchmark analysis. This paragraph reviews the reference you suggested, along with other relevant ones.
>
> ***8. GitHub repository.***
>
> Following your suggestions, we reorganized the GitHub repository and updated the readme with additional information about (1) the data we used, how to obtain it, and how to extend the experiments with custom data; (2) details about the result files and how to produce them. We plan to further improve our repository with the new code for evaluation with AUC.

---

> > ### Comment · Reviewer_N1YT · 2023-08-20
> >
> > I might be mistaken, but I do not believe that ROC AUC calculated with class predictions is equal to balanced accuracy. Do you mean that you instead recalculated balanced accuracy based on the earlier recorded predictions?

---

> > > ### Author Response · Authors · 2023-08-21
> > >
> > > We agree with the reviewer that, in general, ROC AUC with predicted labels and balanced accuracy are not equal.
> > > However, they are the same in our specific case.
> > >
> > > To prove this, first note that scikit-learn's [roc_auc_score](https://scikit-learn.org/stable/modules/generated/sklearn.metrics.roc_auc_score.html#sklearn.metrics.roc_auc_score) computes the ROC AUC for predicted labels from a linear interpolation of the ROC curve [1]. Therefore,
> > > $$ROC\ AUC = \frac{TPR\cdot FPR}{2} + \frac{(1+TPR)\cdot(1-FPR)}{2} = \frac{TPR + (1-FPR)}{2} = \frac{TPR + TNR}{2}$$
> > > which is the balanced accuracy.
> > >
> > > This result is in line with the scikit-learn documentation [2], which states:
> > >
> > > > In the binary case, balanced accuracy is equal to the arithmetic mean of sensitivity (true positive rate) and specificity (true negative rate), or the area under the ROC curve with binary predictions rather than scores.
> > >
> > > ___
> > >
> > > [1] Muschelli III, John. "ROC and AUC with a binary predictor: a potentially misleading metric." *Journal of classification* 37.3 (2020): 696-708.
> > >
> > > [2] https://scikit-learn.org/stable/modules/model_evaluation.html#balanced-accuracy-score, Section 3.3.2.4

---

> > > > ### Comment · Reviewer_N1YT · 2023-08-24
> > > >
> > > > Right, thank you! All concerns are addressed adequately. I don't see a reason for the change of AUC to Balanced Accuracy to affect the score negatively, so I will keep my score as is.

---

### Official Review · Reviewer_siZ7 · 2023-07-24
**A benchmark of categorical encoders for binary classification**

**Rating:** 6
**Confidence:** 3
**Correctness:** Correct.
**Clarity:** Well written. The paper is easy to fo…

**Strengths:**

S1. More datasets (50 datasets) are used than existing benchmarks.

S2. More factors are considered to measure the encoders, the quality metrics, the ML models used, the tuning strategy, the aggregation strategy.

S3. The motivation and advantage are introduced clearly.

**Additional Feedback:**

See O1 and O2.

**Documentation:**

Yes.

**Ethics:**

No.

**Limitations:**

Authors have mentioned the limitations of their work.

**Opportunities For Improvement:**

O1. Descriptions of experimental setting and result analysis are described too simple.

O2. Neural network encoders are not considered. As a comprehensive benchmark, more encoders should be included.

O3. More details about the datasets used should be introduced.

**Relation To Prior Work:**

Yes.

**Summary And Contributions:**

Categorical encoder is an important problem. The paper systematically analyzes the limitation of existing encoder benchmarks and provides a comprehensive benchmark of categorical encoder.

---

> ### Author Response · Authors · 2023-08-11
>
> Thank you for your review. We would ask you to clarify your observations to help us address them.
>
> ### O1.
>
> We find the current comment rather general, as it does not give us any hints on how to improve the paper (which we would be very willing to do).
> Could you please provide more details or specific suggestions?
>
> ### O2.
>
> Unfortunately, we are not sure what exactly you refer to with ``neural network encoders''.
> Neural networks tend to use the same categorical feature encoders for tabular data that we are investigating.
> We would appreciate it if you could clarify this issue as well.
>
> We have examined all encoder types from the existing benchmarks.
> For a detailed list, see the ``Encoder Family'' section in Table 1 of our paper and Section 7.1. of the Supplementary Material.
>
> ### O3.
>
> In line with common scientific practice, this information is part of the supplementary material, as follows:
> Table 11 on Page 16 lists details of each data set in a comprehensive manner.
> It gives names, relevant references, and unique OpenML identifiers.
> This table also provides the number of rows, total attributes, count of categorical attributes, and the highest categorical attribute cardinality for each dataset.
> It also contains which datasets were used for full tuning.
> For further transparency, our GitHub repository includes code that facilitates automatic downloading and preprocessing of these datasets.

---

### Comment · Area_Chair_kFDY · 2023-08-27
**please reply to author replies**

Dear reviewers siZ7 and QEyt,

Would you please check in, to confirm whether you have red the rebuttal of the authors, and whether this has changed the way you rate the paper?

Additionally, reviewer QEyt, can you please still deliver a review, as you committed to by accepting the reviewer assignment?

Best,
The AC

---

### Decision · Program_Chairs · 2023-09-22

**Decision:**

Accept (Poster)

**Comment:**

The authors come up with an extensive benchmark of categorical encoders for binary classification. This is an important problem, where not much research has been done towards.

The average score is misleading: one reviewer did leave a 1 (because the anonymity) and was not available for correcting this mistake.

It is hard to assess the quality of the reviews, since eventually only two reviewers have submitted a valid review. Additionally, one of these is quite shallow, and did not report back to the authors. Therefore, I will mostly base my assessment on the review of reviewer N1YT (who seems to have an excellent track record in benchmarking studies) and my own observations from reading the paper and assessing the benchmark.

While none of the reviewers identified dealbreakers or matters that should be improved (except for the discussion on AUC, which has been resolved) there remain some questions (also raised by reviewer N1YT) about the usability of the repository, and whether other people can easily analyze the results. Why not integrate these results with, for example, OpenML.org (the platform where all datasets came from)? This point was resolved by the remark that the readme will be further improved in the future, but in my opinion this is also something that should have been taken more seriously.

Finally, I would like to comment on the match of this work in the datasets and benchmark track. When looking at the scope of this track, it seems that this part would fall under "Benchmarks on new or existing datasets, as well as benchmarking tools" (on the part: existing). It seems that the authors have compiled a set of existing datasets, and benchmarked a wide range of categorical encoders on it. However, I personally would expected either a) a better integration with existing tools (such as the OpenML benchmark suites), such that other people could build upon this, or b) more clear analysis of the results, with clear conclusions.

Taking all into consideration, I believe that this paper could receive the benefit of the doubt, and can be recommended for a poster presentation at NeurIPS.